# Germination and Simulated Gastrointestinal Digestion of Chickpea (*Cicer arietinum* L.) in Exhibiting In Vitro Antioxidant Activity in Gastrointestinal Epithelial Cells

**DOI:** 10.3390/antiox12051114

**Published:** 2023-05-18

**Authors:** Ashley Newton, Kaustav Majumder

**Affiliations:** 1Department of Food Science and Technology, University of Nebraska-Lincoln, Lincoln, NE 68588-6205, USA; 2256 Food Innovation Center, Nebraska Innovation Campus, Department of Food Science and Technology, University of Nebraska-Lincoln, Lincoln, NE 68588-6205, USA

**Keywords:** chickpea, germination, gastrointestinal digestion, antioxidant activity, bioactive peptides

## Abstract

Plant-based proteins, in particular pulse proteins, have grown in popularity worldwide. Germination, or sprouting, is an effective method to release peptides and other dietary compounds. However, the combination of germination and gastrointestinal digestion in enhancing the release of dietary compounds with potential health-beneficial biological activity has yet to be entirely elucidated. The present study illustrates the impact of germination and gastrointestinal digestion on the release of dietary compounds with antioxidant activity from chickpeas (*Cicer arietinum* L.). Germination up to 3 days (D0 to D3) increased the peptide content by denaturing chickpea storage proteins and increased the degree of hydrolysis (DH) in the gastric phase. The antioxidant activity was measured at three different dosages (10, 50, and 100 μg/mL) and compared between D0 and D3 on human colorectal adenocarcinoma cells (HT-29). A significant increase in antioxidant activity was observed in the D3 germinated samples in all three tested dosages. Further analysis identified 10 peptides and 7 phytochemicals differentially expressed between the D0 and D3 germinated samples. Among the differentially expressed compounds, 3 phytochemicals (2′,4′-dihydroxy-3,4-dimethoxychalcone, isoliquiritigenin 4-methyl ether, and 3-methoxy-4,2′,5′-trihydroxychalcone) and 1 peptide (His-Ala-Lys) were identified only in the D3 samples, indicating their potential contribution towards the observed antioxidant activity.

## 1. Introduction

The interest in plant-based proteins has increased exponentially, particularly pulse proteins [1]. Chickpeas are the world’s third most crucial pulse commodity, as they are consumed in many different cultures globally and are known for their critical nutritional benefits [2,3]. The protein content of chickpeas ranges from 17–30%, and the high amounts of essential amino acids such as lysine and arginine allow pulses to possess many functional properties that provide the opportunity to produce novel health benefits [4]. Compounds such as bioactive peptides, polyphenols, and other minor compounds contribute to the functionality of pulses by exhibiting bioactive effects such as antioxidant, antiproliferative, and anti-inflammatory activities [4,5].

Bioactive peptides are inactive within the parent protein and are released during processing [6]. This processing can include enzymatic hydrolysis, microbial fermentation, or physical processing methods such as germination and cooking [6]. Bioactive peptides’ effect must retain their structure through gastrointestinal (GI) digestion and be absorbed intact. Once they reach the bloodstream, they can exert their bioactive physiological function [7]. Most identified bioactive peptides are between 2–20 amino acids and less than 3 kDa in molecular weight [8].

Oxidative stress, the increased production of reactive oxygen species (ROS), is an increasing factor in the fatality of cardiovascular diseases. An increase in ROS has been linked to low-density lipoprotein (LDL) oxidation and endothelial dysfunction [9]. Dietary antioxidants, such as vitamin C and carotenoids, have been shown to fight against oxidative stress. In the last two decades, interest has leaned towards antioxidative polyphenols and peptides found in dietary substances [10,11]. Previous studies have discovered that edible dry beans can modulate cardiovascular risk factors in an in vivo model [11]. Additionally, there has been a focus on the structure–function relationship of antioxidative peptides regarding the amino acid sequence and related bioactivity [12,13].

Germination is one processing method that produces beneficial bioactive compounds [14]. During the germination process of seeds and pulses, the seed absorbs water, which allows for the activation of the metabolism. Germination activates metabolic processes that produce enzymes and co-factors that increase the nutritive value and reduce the concentration of anti-nutritional factors (phytic acid, tannins, etc.) that are naturally present in the seed or pulse [15,16]. Once the metabolism is activated, a radicle emerges, starting seedling production. Germinated pulses and seeds exhibit many physiological and chemical changes within food, including changes in lipid content in germinated flax seed [17], increases in the water-soluble components in germinated soybean [18], and changes in the dietary fiber contents of germinated dark beans and lentils [19], among others.

Previous studies have used many types of pulses and seeds, but chickpea germination has yet to be further investigated. Furthermore, germination with simulated GI digestion can break down proteins into small peptides with potential bioactivity [20]. Thus, this study aimed to evaluate the efficacy of germination and simulated GI digestion to produce potentially bioactive peptides from chickpeas through an in vitro antioxidant activity assay and identify the active compounds via mass spectrometry analyses.

## 2. Materials and Methods

### 2.1. Materials

The chickpeas were provided by the University of Nebraska-Lincoln Panhandle Research Station in Scottsbluff, NE. The colorectal adenocarcinoma cells (HT-29) (ATCC HTB-38^TM^), McCoy’s 5A Medium (ATCC 30-2007^TM^), and penicillin–streptomycin solution (30-2300^TM^) were purchased from ATCC (Manassas, VA, USA). The Trypsin/EDTA solution (CC-5012) was purchased from Lonza (Basel, Switzerland). Hank’s Balanced Salt Solution (HBSS; 14175095) and fetal bovine serum (FBS; A3160501) were purchased from Thermo Scientific (Waltham, MA, USA). The phytic acid assay kit (K-PHYT) was purchased from Neogen (Lansing, MI, USA). If not stated, the chemicals were reagent grade and purchased from Sigma (St. Louis, MO, USA).

### 2.2. Preparation of the Germinated Chickpeas

The dry chickpeas were soaked in a 2% (*v*/*v*) 0.5 M sodium hypochlorite solution in a 1:2 ratio for 10 min to sterilize and inhibit microbial growth during germination [21]. The chickpeas were rinsed with deionized (DI) water three times, then soaked in 100 mL DI water overnight in the dark at room temperature (22–25 °C). The following day, the chickpeas were drained and rinsed with DI water to begin germination. The chickpeas were germinated for up to five days in germination jars with strainer lids (Kenley Mason Jars with Screen Lids, ASIN: B078TLZ33Q) in the dark at room temperature. Samples were taken every day for up to 5 days, root length (cm) was measured, and germination efficiency was calculated as such:Germination Efficiency (GE %) = # of germinated beans/# of total beans used

Based on root length and GE, the samples taken on days 0 and 3 were used for further analysis (Figure 1). The samples were stored at either −80 °C and lyophilized for protein extraction and anti-nutritional factors analysis or 4 °C for simulated GI digestion.

### 2.3. Protein Extraction and Total Soluble Protein Content

Soluble protein was extracted and used to measure total protein content. Briefly, lyophilized ungerminated or start of germination (day 0; D0) and germinated chickpeas (day 3; D3) were ground into a fine powder. Approximately 30 mg of the powder and 1.5 mL of a 2% NaCl (*w*/*v*) solution were added to 2 mL conical tubes that contained 2.8 mm stainless steel beads (D1033-28; Benchmark Scientific, Sayreville, NJ, USA). The samples were homogenized in a benchtop homogenizer (BS-BEBU-3; Benchmark Scientific, Sayreville, NJ, USA) in 40 s intervals at 4000 rpm. After homogenization, the samples were centrifuged at 10,000× *g* for 10 min. Extraction was performed in triplicate for each germination day. The supernatant was collected and diluted for soluble protein and further analyses.

Lowry’s method was used to measure the total soluble protein content of the extracted supernatant [22]. The samples were assayed in triplicate, adjusted by sample weight, and expressed as g protein/100 g sample.

### 2.4. Total Peptide Content of the Germinated Chickpeas

The Pierce^TM^ Quantitative Fluorometric Peptide Assay (23290; Thermo Scientific, Waltham, MA, USA) was used to quantify the total peptide content in the extracted supernatant of D0 or the D3 chickpea digest. The samples were assayed in triplicate, adjusted by the amount of soluble protein, and expressed as mg peptide/100 g sample.

### 2.5. Sodium Dodecyl Sulfate–Polyacrylamide Gel Electrophoresis (SDS-PAGE)

The effect of germination on the chickpea-extracted protein was assessed by SDS-PAGE analysis. The supernatant was mixed in a 1:1 ratio with 2× Laemmli sample buffer (1610737; Bio-Rad, Hercules, CA, USA) to obtain a 2 mg/mL protein solution and heated at 95 °C for 5 min. Approximately 15 μL of the sample was loaded into wells of a polyacrylamide gel (4568096; Bio-Rad, Hercules, CA, USA) to obtain a 30 μg protein concentration per well. Precision Plus Protein Dual Color Standard was used as a molecular weight marker (1610374; Bio-Rad, Hercules, CA, USA). The gel was run at 80 V, 0.3 A, for 90 min using 10X Tris/Glycine/SDS running buffer (1610732; Bio-Rad, Hercules, CA, USA). The gel was dyed with Coomassie Brilliant Blue R-250 for 20 min and then de-stained with methanol: acetic acid: DI water solution (20:10:70, *v*/*v*/*v*). Gel images were captured with an Odyssey CLx imaging system (Licor, Lincoln, NE, USA; 169 μM resolution, focus offset of 0.0 mm, lowest scan quality).

### 2.6. Phytic Acid Determination

The total phytic acid content of the ground germinated chickpeas was measured using a Megazyme phytic acid/total phosphorus assay kit (K-PHYT, Neogen, Lansing, MI, USA). The samples were assayed in triplicate.

### 2.7. Simulated Gastrointestinal Digestion

The simulated gastrointestinal digestions were performed in triplicate, following the INFOGEST protocol with slight modifications as per our previous study [23,24]. The simulated salivary, gastric, and intestinal fluids were prepared as detailed in Minekus et al., 2014 [25]. Whole germinated chickpeas (D0 and D3) were mashed with a mortar and pestle to mimic mechanical chewing. The oral, gastric, and intestinal phases were marked with a change in pH (7, 3, and 7, respectively). The end of the intestinal phase was marked by adding 8 mL of trypsin–chymotrypsin inhibitor. The solution was centrifuged at 10,000× *g* for 20 min to obtain digested (supernatant) and undigested (precipitate) fractions. Part of the supernatant was subjected to <3 kDa ultra-filtration via Amicon^®^ Stirred Cell 200 mL (UFSC20001; Sigma, St. Louis, MO, USA) to obtain a filtrate (<3 kDa MW) and retentate (>3 kDa MW). All digested, undigested, filtrate, and retentate fractions were frozen at −80 °C and lyophilized. The degree of hydrolysis was determined by the pH-stat method equations [24,26]. The <3 kDa germinated chickpea-filtrate digest (CD), defined hereafter as either D0-CD or D3-CD, was used for further analysis.

### 2.8. Cell Culture

Human colorectal adenocarcinoma cells (HT-29) (HTB-38^TM^; ATCC^®^, Manassas, VA, USA) were grown in McCoy’s 5A Medium (30-2007^TM^; ATCC^®^, Manassas, VA, USA) supplemented with a 0.5% penicillin–streptomycin cocktail (*v*/*v*) and 10% fetal bovine serum (*v*/*v*) at 37 °C and 5% CO_2_ in humidified conditions. The cell media were replaced every 2 days, and the cells were grown for 3–4 days to reach 90% confluency and were used for the experiments. Cells between passages 4–8 were used for this study.

### 2.9. In Vitro Antioxidant Activity

The antioxidant activity of D0-CD and D3-CD was determined according to the cellular antioxidant assay with minor modifications [27,28]. Briefly, HT-29 cells were seeded in black-sided clear-bottom 96-well microplates at 20,000 cells/well density. After reaching 90% confluency, the cells were washed twice with HBSS tempered at 37 °C and incubated with synchronization media, consisting of McCoy’s 5A media supplemented with a 1% penicillin–streptomycin (*v*/*v*) cocktail and 1% fetal bovine serum (FBS) (*v*/*v*) for 2 h at 37 °C. After synchronization, the cells were incubated with 100 μL (D0-CD or D3-CD at 10, 50, and 100 μg/mL dissolved in the synchronization media) treatment solution. The cells were incubated at 37 °C and 5% CO_2_ for 150 min. After this incubation, 50 μL of 50 mM diacetyldichlorofluorescein (DCFH-DA) was added to the cells, and they were incubated for 30 min at 37 °C and 5% CO_2_. The cells were washed twice with tempered HBSS and replaced with 50 μL of HBSS. 2′2′-azobis(2-methylpropionamidine) dihydrochloride (AAPH) was used as the oxidizer at a concentration of 600 μM per well, and one set of cells was only treated with AAPH only, labeled as the positive control (PosC). Approximately 50 μL of AAPH solution was added to each well in a total volume of 100 µL/well. Quercetin (20 μM) (Millipore Sigma, Q4951, St. Louis, MO, USA) was used as an inhibitor control in this assay, and one set of cells was left untreated (no AAPH, no inhibitor, and no treatment), labeled as the negative control (NegC). Fluorescence was immediately measured using excitation and emission wavelengths of 485 and 528 nm at 1 min intervals for 60 min. Each treatment group’s relative fluorescence unit (RFU) was plotted against the time. The area under the curve (AUC) was calculated for all the treatment groups and they were compared against each other.

Furthermore, cellular antioxidant activity (%) was calculated using the below equation:(1)CAA unit(%)=100−(∫SA/∫CA)×100
∫*SA* is the integrated area under the sample curve, and ∫*CA* is the integrated area under the control curve. The AUC for each sample was used as ∫*SA* for the *CAA* analysis, and the AUC of the PosC curves was taken as ∫*CA*.

### 2.10. Identification of Free Amino Acids

For free amino acids analysis, 50 mg of the D0-CD and D3-CD samples was extracted using liquid–liquid biphasic extraction with distilled water, methanol, and chloroform solution, as described previously [29]. Free amino acids in the samples were derivatized using AccQ TAG Ultra Reagent (186003836; Waters Corp., Milford, MA, USA) and analyzed using high-performance liquid chromatography (HPLC) and coupled with a UV detector at 254 nm. Standard curves were run alongside the samples to calculate the concentration of amino acids in the samples from the peak areas detected. The analyses were performed in triplicate, and the results were expressed as μg/g sample.

### 2.11. Small Compound Identification through Untargeted HILIC LC-MS/MS

The small molecular weight compound profiles from D0-CD and D3-CD were analyzed as described in our previous studies, with minor modifications [24,30]. Briefly, the lyophilized samples were resuspended in LC-MS/MS-grade water to a concentration of 20 μg/μL. The samples were diluted twice with 100% acetonitrile for a 50 μg injection. The injection was loaded onto an XBridge Amide 3.5 μm column (186004868; Waters Corp., Milford, MA, USA) using a Vanquish HPLC (Thermo Fisher, Waltham, MA, USA) at 45 °C and a flow rate of 400 μL/min with gradients A and B [24]. The phases went as follows: 90% B to 30% B in 13 min and then back to 90% B in 0.5 min. The data were acquired on a QE-HF mass spectrometer (Thermo Fisher, Waltham, MA, USA) using positive ion mode, with a mass range of 60 to 900 *m*/*z* on single charge ions at 60,000 resolutions with an AGC target of 3 × 10^6^ and a maximum ion time of 50 ms. The ions were further fragmented by HCD using an isolation window of 1.6 *m*/*z* at 15,000 resolutions.

Hydrophilic interaction chromatography (HILIC) separation data were analyzed for small compound quantification and identification using Progenesis QI (v. 2.4; Waters Corp., Milford, MA, USA). Peak abundance was normalized for differences in sample loading by a total ion chromatogram. NIST MS/MS v. 1.0 was used for the MS/MS library search, and compound identification was filtered using a mass accuracy of <5 ppm and an isotopic similarity of at least 90%. The MS2 spectra were manually reviewed to match the MS2 spectra from the database.

### 2.12. Statistical Analysis

The data sets were confirmed to be normally distributed using the Shapiro–Wilk test. If the data were normally distributed, statistical analysis was performed for multiple groups using a one-way analysis of variance (ANOVA), followed by Tukey’s multiple comparisons test. Otherwise, the non-parametric Kruskal–Wallis test was used, followed by Dunn’s multiple comparisons test. Where appropriate, an unpaired two-tailed *t*-test was used to compare the data between the D0 and D3 samples. All of the statistical analyses were performed using GraphPad Prism software (v. 9, San Diego, CA, USA). The data are presented as the ±standard deviation (SD), and statistical significance is considered at *p* < 0.05.

## 3. Results

### 3.1. Whole Germinated Chickpea Nutritional Analysis

Chickpeas germinating for up to 5 days, with their root length and germination efficiency measured each day, statistically increased from days 1 to days 3, 4, and 5 (*p* < 0.01) for root length (Figure 1A) but only showed a significant difference from day 1 to day 3 (*p* < 0.01) for germination efficiency, as shown in Figure 1B. There was no significant difference between days 3, 4, and 5 in root length and GE; therefore, chickpeas at the start of germination or ungerminated (D0) and germinated (D3) were chosen for further analysis.

As shown in Figure 2, the total soluble protein and total peptide content significantly increased during germination. The D0 germinated chickpeas had a total soluble protein content of 16.36 g protein/100 g sample compared to D3 at 17.19 g protein/100 g sample (*p* < 0.05) (Figure 2A). The total soluble peptide content (Figure 2B) significantly increased from D0 to D3 of germination, from 0.187 mg peptide/100 g sample to 0.941 mg peptide/100 g sample, respectively (*p* < 0.0001). The total phytic acid content decreased significantly (Figure 2C) between D0 and D3 (*p* < 0.01) due to the release of enzymes during germination that can break down large compounds. Furthermore, after 3 days of germination, lipoxygenase at 90 kDa completely disappeared, while the legumin subunits at 47 kDa and the vicilin subunits at 33 and 35 kDa denatured and decreased in size (Figure 2D). Bands at 10 and 15 kDa were identified as albumin and 7S vicilin subunits, respectively, and no change was observed in these proteins.

### 3.2. Proteolytic Activity of the Germinated and GI-Digested Chickpeas

The degree of hydrolysis (DH) was measured and calculated through the pH-stat method [24]. The gastric phase (Figure 3A) and intestinal phase (Figure 3B) were measured separately. There was a significant increase in the gastric DH (*p* < 0.05) from D0 (6.16 ± 0.80) to D3 (9.08 ± 0.97). In contrast, a significant decrease was observed in the intestinal DH (*p* < 0.01) from D0 (3.54 ± 0.20) to D3 (2.43 ± 0.21). The total peptide content significantly increased between D0-CD and D3-CD, as shown in Figure 3C (*p* < 0.05). This is consistent with the total peptide content before GI digestion, which significantly increased as the germination time increased.

### 3.3. In Vitro Antioxidant Activity of D0-CD and D3-CD

Three dosages (10, 50, and 100 μg/mL) were used to determine the antioxidant activity of the D0-CD and D3-CD samples. After comparing the mean RFU of each treatment group with the PosC group, it was observed that the inhibitor and NegC groups were significantly different (*p* < 0.0001); however, among the treatment groups, only the D3-CD 100 μg/mL group was significantly different (*p* < 0.05) from the PosC group (Figure 4A). While comparing the AUC over 60 min, it was observed that, in the D0-CD samples, there was a significant decrease between the PosC group with the 50 and 100 μg/mL groups (*p* < 0.01 and *p* <0.001, respectively) but not with the lowest dosage of 10 μg/mL (Figure 4B). However, the D3-CD samples showed a significant decrease at the lowest dosage of 10 μg/mL (*p* < 0.01), as well as the two other dosages of 50 and 100 μg/mL (*p* < 0.0001) (Figure 4B). The cellular antioxidant activity (CAA; %) was calculated to compare the most effective dosages among the treatment groups. The inhibitor group showed the highest CAA % with 85.9 ± 3.2, and among the treatment groups, D3-CD 50 and the 100 μg/mL groups showed the highest CAA % at 35.2 ± 17.0 and 38.0 ± 14.2, respectively.

### 3.4. Free Amino Acid and Small Molecule Identification of GCD

Analysis of free amino acids in the D0-CD and D3-CD samples revealed no change in the concentrations of any identified free amino acids. Appendix A summarizes the free amino acids identified and the concentrations measured in these samples. However, further analysis of small molecules using HILIC LC-MS/MS identified and quantified eight polyphenolic compounds (Table 1). Seven compounds identified significantly increased concentration during germination (*p* < 0.05). Notably, three compounds (2′,4′-dihydroxy-3,4-dimethoxychalcone, isoliquiritigenin 4-methyl ether, and 3-methoxy-4,2′,5′-trihydroxychalcone) were only identified in D3-CD, but not in the D0-CD samples.

### 3.5. Identification of Peptides in D0-CD and D3-CD

Appendix A lists the identified di- and tripeptides and their respective measured intensity in the D0-CD and D3-CD samples. Overall, 62 and 63 peptides were identified in D0-CD and D3-CD, respectively; the intensity and distribution of all these peptides are illustrated in Appendix A. Of all the identified peptides, the differential abundance (DA) of 10 peptides (6 di- and 4 tripeptides) was significantly different between the D0-CD and D3-CD samples (Figure 5). Of the 6 DA dipeptides, Gly-Asp (GD), Pro-Asn (PN), Phe-Glu (FE), and Leu-Gln (LQ) were significantly increased in D3-CD compared to the D0-CD samples, whereas Glu-Phe (EF) and Glu-Tyr (EY) were significantly decreased in the D3-CD samples. Among the 4 DA tripeptides, Pro-Val-Lys (PVK), Asp-Lys-Lys (DKK), and Ala-His-Arg (AHR) were markedly reduced in the D3-CD samples, and the tripeptide, His-Ala-Lys (HAK) was only identified in the D3-CD samples, not in the D0-CD samples (Figure 5).

## 4. Discussion

This study demonstrates that germination and simulated GI digestion could modulate chickpeas’ biological activity and exhibit an in vitro antioxidant effect. The combination of germination and GI digestion produced small compounds and peptides that could contribute to the observed antioxidant activity. Germination has been shown to modulate the nutritive value of pulses by releasing endogenous enzymes to hydrolyze large compounds and decrease anti-nutritional factor concentration, thus increasing protein digestibility [14]. The present study establishes the first step in measuring the biological activity of germinated and GI-digested chickpeas and identifying potential bioactive dietary compounds.

The total soluble protein content increased in the D3 chickpea samples, which is similar to an earlier study that showed a significant increase in the crude protein content of chickpeas after 3 days of germination [31]. Other studies have identified increased protein content after the germination of cowpea [32] and soybean [33,34]. However, there have been concerns about overusing chemical assays, such as Lowry’s assay, in determining the accurate concentration of total protein in proteinaceous food samples. Previous studies have found that Lowry’s protein determination method is photosensitive, resulting in total protein discrepancies of up to 10% [35,36]. Furthermore, studies have been carried out to compare the total protein analytical methods and quantification of total protein, where the inaccuracies were linked to interfering substances and the overestimation of total protein concentration [36]. Therefore, the increase in the total protein content of the germinated chickpea samples could be attributed to overestimation. Thus, the present study also measures the peptide content to address any issues related to inaccuracies. As expected, the total peptide content of the chickpeas significantly increased over 3 days of germination in this study, suggesting that germination activates proteases and enhances proteolytic actions to cleave the storage proteins in chickpeas. Further studies should utilize various total protein analytical methods to identify the total protein content of germinated chickpeas and analyze the total nutritional composition of chickpeas germinated for up to 3 days. Anti-nutritional factors, such as phytic acid, are highly abundant in various legumes [37]. This study measured phytic acid content as a single anti-nutritional factor expected to decrease during germination. A significant decrease in total phytic acid content was observed, suggesting that germination could reduce the abundance of anti-nutrients in legumes. However, further investigation on the whole profile of anti-nutritional factors in pulses during germination and other compounds, such as trypsin inhibitors and tannins, should be measured to strengthen this claim.

After D3 of germination, lipoxygenase (90 kDa MW) completely disappeared, while 11S legumin subunits (47 kDa MW) and 7S vicilin subunits (33, 35 kDa MW) denatured. Protein fractions such as legumins and vicilins are part of the globulin storage protein group, the major protein fraction of chickpeas [38,39]. The results indicate that the proteolytic effect of germination predominantly affects higher MW proteins such as lipoxygenase and has little impact on globulin-type storage proteins.

The present study shows that the DH is significantly increased in the gastric phase and then decreased in the intestinal phase from days 0 to 3 of germination. Trypsin is the primary enzyme in intestinal digestion, resulting in a decrease in the intestinal DH after germination, indicating that the proteolytic cleavages of germination may be similar to the tryptic digest sites. Overall, the average DH of the germinated chickpeas increased, with an average of 5.76% in the D3 samples. However, the DH for the germinated chickpeas obtained in this present study is much lower than the DH reported in earlier studies. Germinated *Phaseolus lunatus* seeds were enzymatically hydrolyzed with digestive enzymes for 3 h with a DH of 29.28% [39], possibly due to using different substrates (different pulses have different digestibility), ratios of the GI enzymes, or the method of DH determination. Another study measured the DH of chickpea protein isolates produced from chickpeas germinated for 48 h. They found that the DH reached a high of 1.89 ± 0.47% [40], which more closely matches the results found in the current study. The significant increase in the total peptide content of the D3-CD samples and the DH in the gastric phase suggests that proteolytic activity and protein digestibility occur during GI digestion, emphasizing that the combination of germination and simulated GI digestion can produce smaller peptides, which could exert biological activities.

The combination of germination and enzymatic digestion has been shown to produce biologically active fractions of pulses, including dipeptidyl peptidase-IV (DPP-IV) and α-glucosidase inhibitory activities [41], angiotensin-converting enzyme (ACE) inhibitory activity [20], and antioxidant activity [42]. However, germination and GI digestion have not been elucidated in producing antioxidative peptides from pulses. The results indicated that D3-CD exhibited antioxidant activity against the PosC group across all the tested dosages; however, for the D0-CD samples, the antioxidant activity was only observed in the high dosages, suggesting that germination produces peptides or other compounds which are primarily responsible for the observed antioxidant activity. Cowpeas germinated for up to 48 h and hydrolyzed with Alcalase exhibited antioxidant activity [32]. Germination of jack bean and mucuna bean has been shown to generate antioxidant activity, primarily due to the increase in phenols and catechins [43]. Many studies have identified the antioxidant activity of germinated and enzymatically digested pulses using chemical-based assays; very few have researched the use of in vitro cell-based assays to analyze antioxidant activity. Thus, this study adds to the knowledge of the synergistic effect of germination and GI digestion in producing bioactive dietary compounds for exhibiting antioxidant activity. We used human HT-29 cells in this study as our cell model; as these cells are able to express characteristics of mature intestinal cells, such as enterocytes or mucus-producing cells, this cell line has been used before for similar studies [44]. Additionally, we obtained a stable and reproducible signal for our antioxidant assay while using this cell line rather than Caco2 cell lines, commonly used for this purpose [11]. Future studies should incorporate multiple cell lines to evaluate the actual efficacy of the germinated chickpea digest.

Determining the potential active biomolecules contributing to exhibiting the antioxidant activities in the D3 germinated samples is critical. Thus, the present study analyzed free amino acids, phytochemicals, and small peptides in the D0-CD and D3-CD samples. As observed, there were no significant differences in free amino acid composition, but some interesting changes were observed in the abundance of phenolic compounds and peptides. Seven identified phenolic compounds increased significantly after germination (Table 1). Thus, the perceived antioxidant activity could be attributed to the small phenolic compounds identified in D3-CD that increased significantly. Germination of various legumes increased the total phenolic content with a concomitant rise in antioxidant activity [43]. The germinated chickpeas contained high phenolic and flavonoid contents [18] and increased chemical antioxidant activity [45]. Future work must confirm the antioxidant capacity of the three (2′,4′-dihydroxy-3,4-dimethoxychalcone, isoliquiritigenin 4-methyl ether, and 3-methoxy-4,2′,5′-trihydroxychalcone) polyphenolic compounds which were only expressed in the D3-CD samples.

Ten DA peptides (six di- and four tripeptides) were identified between the D0-CD and D3-CD samples. The 4 DA dipeptides, Gly-Asp, Pro-Asn, Phe-Glu, and Leu-Gln, significantly increased in the D3-CD samples, suggesting their potential role in exhibiting antioxidant activities. Earlier studies identified that the presence of Leu, Ile, or Phe at the amino terminus of di- and tripeptides was responsible for the inhibition of lipid oxidation in a liposomal model system [46], and the presence of Pro could contribute to ferric reducing antioxidant power [47,48]. Interestingly the expression of the peptide Phe-Glu increased, and Glu-Phe decreased in the D3-CD samples, indicating the critical role of amino acids in specific positions in exhibiting biological activities. Further research is required to understand the molecular mechanisms of action. Among the four DA tripeptides, the expression of three peptides, Pro-Val-Lys, Asp-Lys-Lys, and Ala-His-Arg, significantly decreased in the D3-CD samples, but one tripeptide, His-Ala-Lys, was only detected in the D3-CD samples. Notably, the presence of a histidine (His) residue in a tripeptide was found to inhibit lipid oxidation [49]. Thus, all four DA peptides, Gly-Asp, Pro-Asn, Phe-Glu, Leu-Gln, and His-Ala-Lys, could be responsible for exhibiting antioxidant activity, which is the product of germination and GI digestion.

## 5. Conclusions

In summary, three days of germination were shown to modulate the protein, peptide, and phytic acid content of chickpeas. The combination of germination and GI digestion increased the total peptide content and the gastric degree of hydrolysis (DH). The D3-CD samples exhibited in vitro antioxidant activity for all dosages as measured via HT-29 cells. Three polyphenolic compounds (2′,4′-dihydroxy-3,4-dimethoxychalcone, isoliquiritigenin 4-methyl ether, and 3-methoxy-4,2′,5′-trihydroxychalcone) and one tripeptide (His-Ala-Lys) which were only identified in the D3-CD samples could potentially be responsible for the observed antioxidant effect in the HT-29 cells.

## Figures and Tables

**Figure 1 antioxidants-12-01114-f001:**
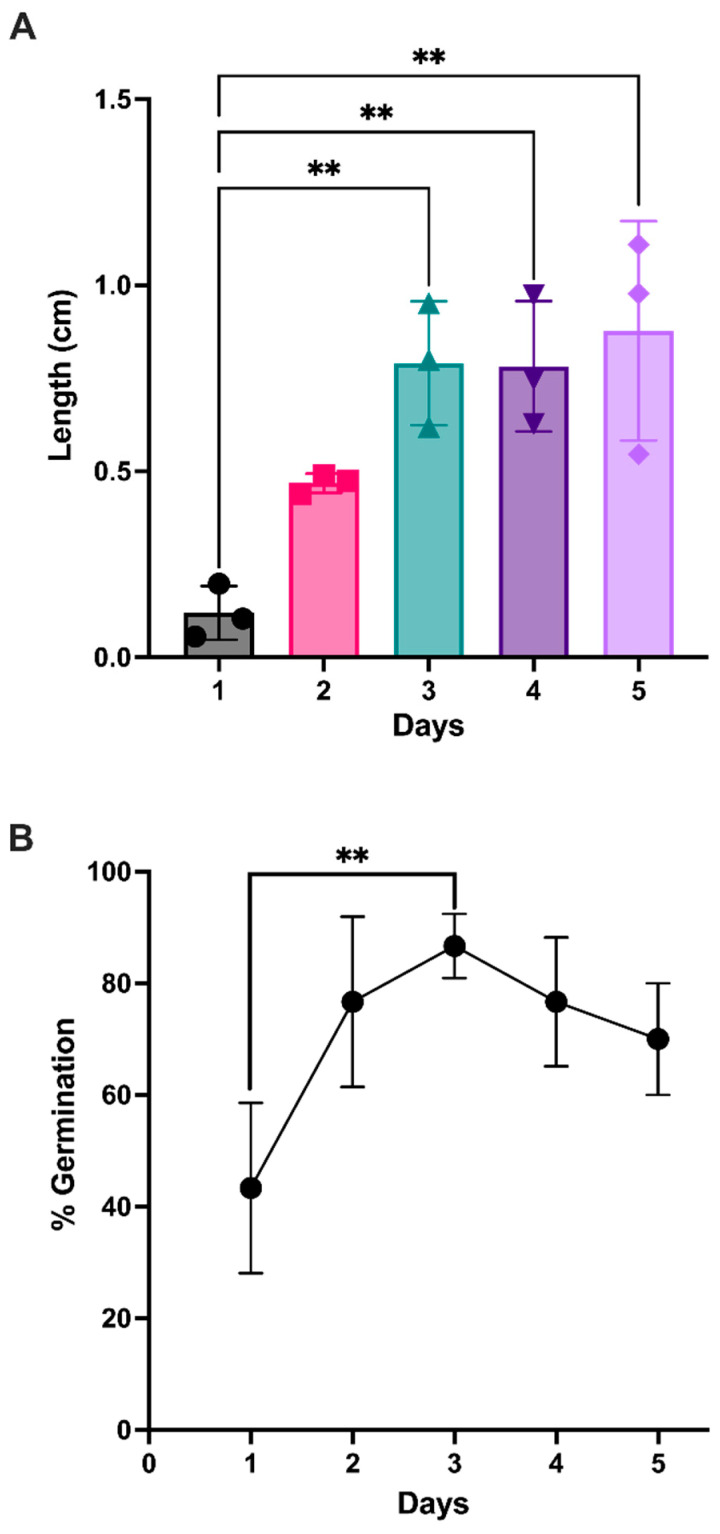
(**A**) Root length (cm) of the germinated chickpeas and (**B**) the germination efficiency of the chickpeas for up to 5 days. Error bars represent the standard deviation (SD); ** represents a statistically significant difference (*p* < 0.01; one-way ANOVA and Tukey’s multiple comparisons test; *n* = 3).

**Figure 2 antioxidants-12-01114-f002:**
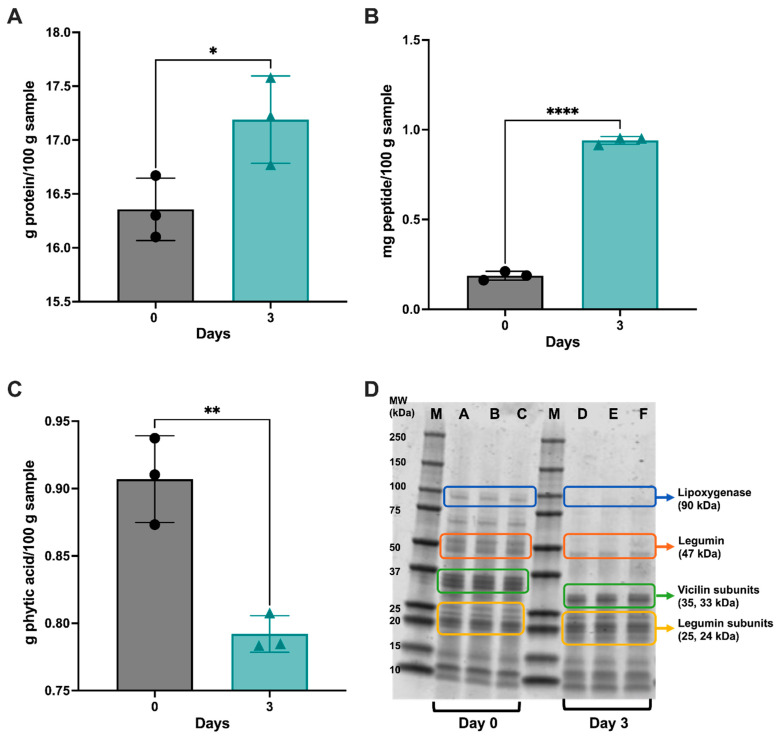
(**A**) Total soluble protein content (g protein/100 g sample), (**B**) total peptide content (mg peptide/100 g sample), and (**C**) total phytic acid content (g phytic acid/100 g sample) of the germinated chickpeas. Error bars represent the standard deviation (SD); *, **, and **** represent statistically significant differences (*p* < 0.05, *p* < 0.01, and *p* < 0.0001, respectively; two-tailed unpaired *t*-test; *n* = 3). (**D**) Sodium dodecyl sulfate–polyacrylamide gel electrophoresis (SDS-PAGE) of the germinated chickpeas (M = protein molecular weight marker; A, B, and C = day 0 (D0); D, E, and F = day 3 (D3) samples).

**Figure 3 antioxidants-12-01114-f003:**
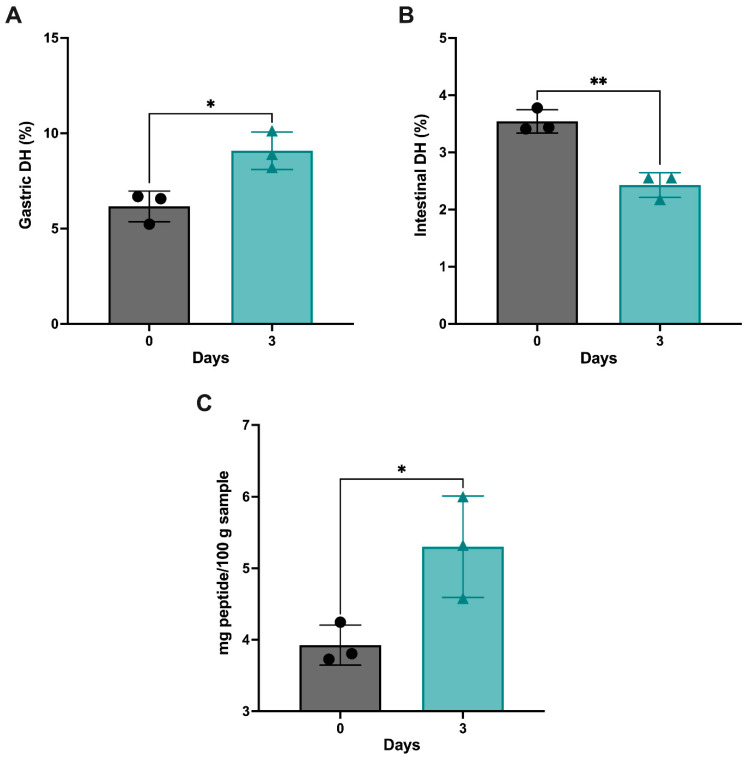
(**A**) Gastric degree of hydrolysis (DH), (**B**) intestinal DH, and (**C**) total peptide content (mg peptide/100 g sample) of the germinated chickpeas after GI digestion. Error bars represent the standard deviation (SD); * and ** represent statistically significant differences (*p* < 0.05 and *p* < 0.01, respectively; two-tailed unpaired *t*-test; *n* = 3).

**Figure 4 antioxidants-12-01114-f004:**
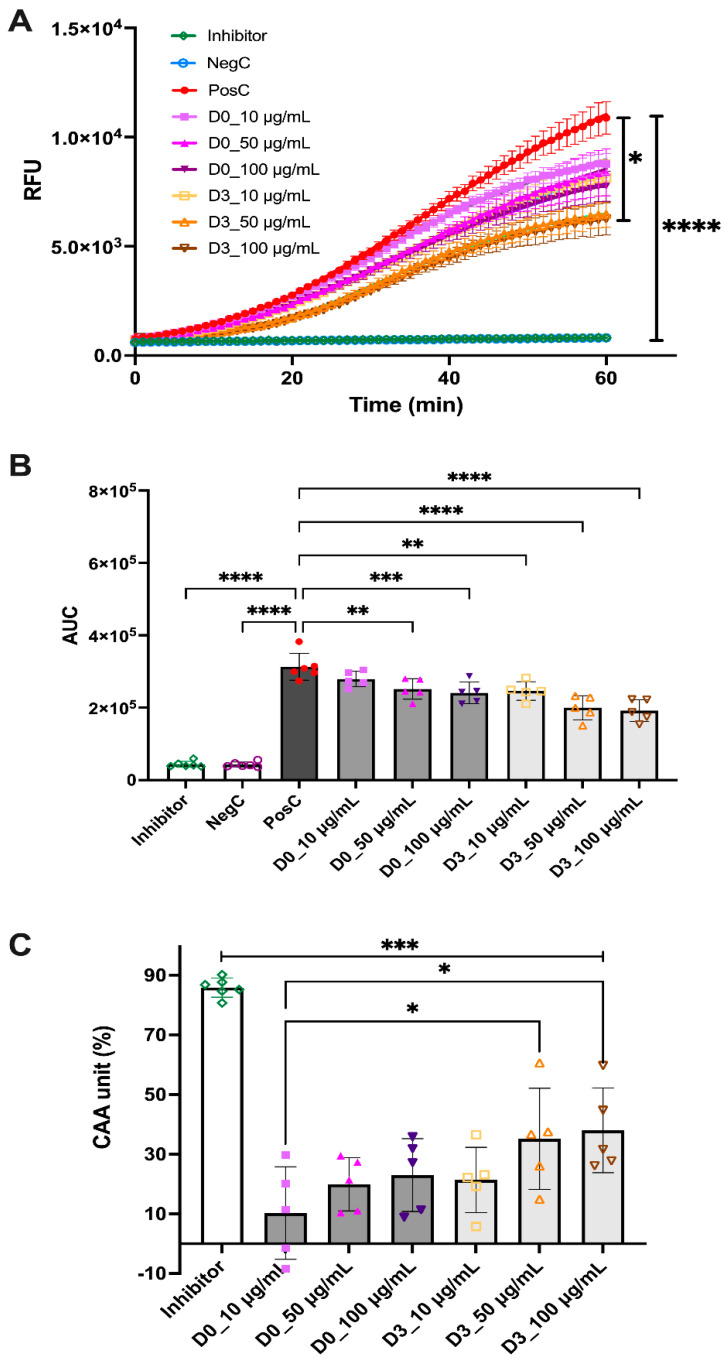
(**A**) Relative fluorescence unit (RFU) to measure the antioxidant activity minutes, (**B**) area under the curve (AUC) for all treatment groups measuring the antioxidant activity over 60 min, and (**C**) cellular antioxidant activity (CAA) for all the groups. Error bars represent the standard deviation (SD); *, **, ***, and **** represent statistically significant differences (*p* < 0.05, *p* < 0.01, *p* < 0.001, and *p* < 0.0001, respectively; Kruskal–Wallis test with Dunn’s multiple comparisons test in (**A**). One-way ANOVA with Tukey’s multiple comparisons test in (**B**,**C**): *n* = 5–6; for the cell culture study, each replication (*n* = 3) from the previous experiments was measured twice (*n* = 3 × 2 = 6) and treated as independent; some datasets were lost during the assay due to technical issues.

**Figure 5 antioxidants-12-01114-f005:**
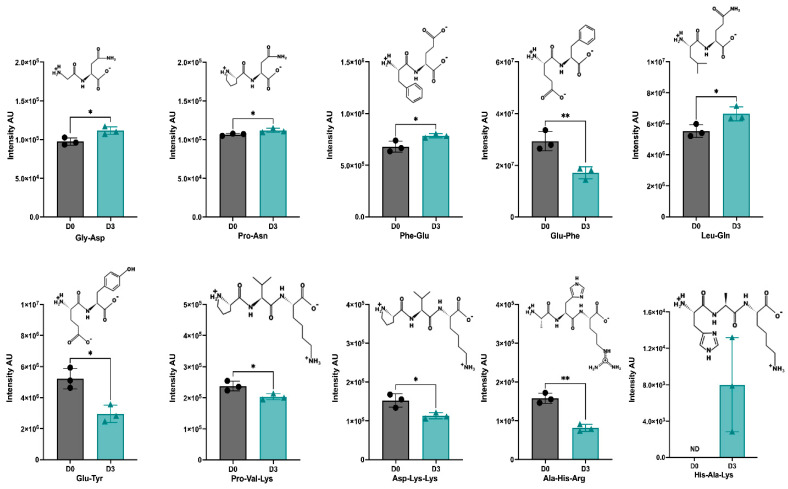
Ten peptides identified in D0-CD and D3-CD with differential abundances; differences in the intensities and structures are illustrated for specific sequences. Error bars represent the standard deviation (SD); * and ** represent statistically significant differences (*p* < 0.05 and *p* < 0.01, respectively; two-tailed unpaired *t*-test; *n* = 3).

**Table 1 antioxidants-12-01114-t001:** List of identified polyphenolic compounds and their intensity, *m*/*z*, and retention time (min).

Compound	Intensity of Day 0 (D0-CD)	Intensity of Day 3 (D3-CD)	*m*/*z*	Retention Time (min)
*o*-Coumaric acid ^#^	1,633,790 ± 63,865	1,509,267 ± 35,909	165.0546	8.91
Isoliqueritigenin *	290 ± 118	12,898 ± 1233	257.0807	2.96
3-Hydroxy-3′-methoxyflavone *	57 ± 50	3086 ± 950	269.0805	4.38
Isodalbergin *	16,751 ± 7693	4,670,146 ± 1,343,553	269.0807	2.84
Isoliquiritigenin 4-methyl ether *	0	6226 ± 779	271.0964	2.73
3-Methoxy-4,2′,5′-trihydroxychalcone *	0	13,566 ± 746	287.0911	2.92
Geraldol *	610 ± 712	32,031 ± 1400	301.0705	2.89
2′,4′-Dihydroxy-3,4-dimethoxychalcone *	0	12,082 ± 2568	301.1070	2.75

^#^ Intensity decreased significantly (*p* < 0.05) from day 0 (D0) to day 3 (D3) of germination. * Intensity increased significantly (*p* < 0.05) from day 0 (D0) to day 3 (D3) of germination.

## Data Availability

All of the data is contained within the article and the Appendix A.

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
