# Peer review of "Germination and Simulated Gastrointestinal Digestion of Chickpea (Cicer arietinum L.) in Exhibiting In Vitro Antioxidant Activity in Gastrointestinal Epithelial Cells"

_antioxidants, 2023, doi:10.3390/antiox12051114_

Round 1

Reviewer 1 Report

In the Manuscript " Germination and Simulated Gastrointestinal Digestion of  Chickpea (Cicer arietinum L.) in Exhibiting in-vitro Antioxidant Activity in Gastrointestinal Epithelial Cells" the aim of the authors was to determine whether the impact of germination and gastrointestinal digestion on the release of dietary compounds with antioxidant activity from chickpeas (Cicer arietinum L.). The paper is a potentially interesting manuscript that necessity of same integrations.

1)      The authors should follow the antioxidant activity of the Chickpeas germinating (CG) and GI samples also in a confluent Caco-2 cellular system, that is like to enterocyte, in which oxidative stress has been induced.

2)      It would also be interesting to show whether the antioxidant effect of CG and GI extracts also influenced the adjacent cell-cell junctions in vitro.

Author Response

In the Manuscript " Germination and Simulated Gastrointestinal Digestion of Chickpea (Cicer arietinum L.) in Exhibiting in-vitro Antioxidant Activity in Gastrointestinal Epithelial Cells" the aim of the authors was to determine whether the impact of germination and gastrointestinal digestion on the release of dietary compounds with antioxidant activity from chickpeas (Cicer arietinum L.). The paper is a potentially interesting manuscript that necessity of same integrations.

The authors thank the reviewer for the encouraging comments; we have now addressed all the concerns.

  1. The authors should follow the antioxidant activity of the Chickpeas germinating (CG) and GI samples also in a confluent Caco-2 cellular system, that is like to enterocyte, in which oxidative stress has been induced.

Interesting thought, we aimed to perform the assay on the Caco2 cells, but the signals of the antioxidant assay were stable and strong when we used the HT-29 cells, which is why we have used HT-29 cells for stable and reproducible results. Both HT-29 and Caco2 are immortal colonic epithelial cells, and thus physiologically, they are very similar. In our future studies, we aim to use other regular gastrointestinal cells to see the broad impact of our germinated chickpea.

  1. It would also be interesting to show whether the antioxidant effect of CG and GI extracts also influenced the adjacent cell-cell junctions in vitro.

Thank you for the comment; in our future study, we are focusing on cell-cell junctions and protein expression after our treatments; adding those data will require additional time and resources, which is out of the scope of this manuscript.

Reviewer 2 Report

The present study illustrates the impact of germination and gastrointestinal digestion on the release of dietary compounds with antioxidant activity from chickpeas (Cicer arietinum L.). Germination up to 3 days (D0 to D3) increased the peptide content by denaturing chickpeas storage proteins and increased the degree of hydrolysis (DH) in the gastric phase. The antioxidant activity was measured at three different dosages (10, 50, and 100 mg/mL) and compared between D0 and D3 on Human colorectal adenocarcinoma cells (HT-29). Three phytochemicals (2’,4’-dihydroxy-3,4-dimethoxychalcone, isoliquiritigenin 4-methyl ether, and 3-methoxy-4,2’,5’-trihydroxychalcone) and one peptide (His-Ala-Lys) were identified only in D3 sample, which can be attributed to their antioxidant activity.

To the article, I have next comments and recommendations:

·         Between the number and degrees of Celsius should be a gap, like 25 °C, -80 °C, etc. Correct thoroughly through article.

·         In some used devices, equipment and chemicals should be completed the manufacturer, town, and country, e.g., Kenley Mason Jars with Screen Lids, ASIN: B078TLZ33Q; Odyssey CLx imaging system; Quercetin (20 μM) (Millipore Sigma, Q4951; the GraphPad Prism software (v. 9).

·         Correct: ¦SA is the integrated area under the sample curve, and ¦CA is the integrated area under the control curve. AUC for each sample was used as ¦SA for CAA analysis, and the AUC of PosC curves was taken as ¦CA.

·         Table 1: o-Coumaric acid o should be in italics.

·         References: the references should be unified in the same style: reference 1: 2019; reference 2: 2019; reference 3: 2017; reference 9: correct and complete ISBN; reference 11: Phaseolus vulga L.; reference 31: Cicer arientinum L. and abbreviation of journal; reference 32: Vigna unguiulata L. and abbreviation of journal;  reference 37: instead the full name the journal use its abbreviation; reference 37: abbreviation of journal; reference 40: abbreviation of journal.

·         In Table S2: should be Pro-Glu-Val and the same in Fig. S1; and in addition in Table S2: Pro-Ile and not I Pro-Ile.

Author Response

The present study illustrates the impact of germination and gastrointestinal digestion on the release of dietary compounds with antioxidant activity from chickpeas (Cicer arietinum L.). Germination up to 3 days (D0 to D3) increased the peptide content by denaturing chickpeas storage proteins and increased the degree of hydrolysis (DH) in the gastric phase. The antioxidant activity was measured at three different dosages (10, 50, and 100 mg/mL) and compared between D0 and D3 on Human colorectal adenocarcinoma cells (HT-29). Three phytochemicals (2’,4’-dihydroxy-3,4-dimethoxychalcone, isoliquiritigenin 4-methyl ether, and 3-methoxy-4,2’,5’-trihydroxychalcone) and one peptide (His-Ala-Lys) were identified only in D3 sample, which can be attributed to their antioxidant activity.

To the article, I have next comments and recommendations:

  • Between the number and degrees of Celsius should be a gap, like 25 °C, -80 °C, etc. Correct thoroughly through article.

Corrected as per suggestion.

  • In some used devices, equipment and chemicals should be completed the manufacturer, town, and country, e.g., Kenley Mason Jars with Screen Lids, ASIN: B078TLZ33Q; Odyssey CLx imaging system; Quercetin (20 μM) (Millipore Sigma, Q4951; the GraphPad Prism software (v. 9).

Corrected as per suggestion, wherever possible.

  • Correct: ¦SAis the integrated area under the sample curve, and ¦CA is the integrated area under the control curve. AUC for each sample was used as ¦SA for CAA analysis, and the AUC of PosC curves was taken as ¦CA.

Corrected as per suggestion.

  • Table 1: o-Coumaric acid oshould be in italics.

Corrected as per suggestion.

  • References:the references should be unified in the same style: reference 1: 2019; reference 2: 2019; reference 3: 2017; reference 9: correct and complete ISBN; reference 11: Phaseolus vulga L.; reference 31: Cicer arientinum L. and abbreviation of journal; reference 32: Vigna unguiulata L. and abbreviation of journal;  reference 37: instead the full name the journal use its abbreviation; reference 37: abbreviation of journal; reference 40: abbreviation of journal.

Corrected as per suggestion.

  • In Table S2: should be Pro-Glu-Val and the same in Fig. S1; and in addition in Table S2: Pro-Ile and not I Pro-Ile.

Pyro-Glu-Val is correct in Table S2, and the DA of this peptide is not significantly modulated so it is not in Fig 1.

Reviewer 3 Report

The authors present a study about the effect of germination on the gastrointestinal digestion and antioxidant activity of chickpeas. 

The manuscript is well organized, and the results are interesting. The study is potentially helpful for the food industry and consumers.

1. Does soaking the dried chickpeas in a hypochlorite solution not affect the germination? Shouldn't the chickpeas be rehydrated before sterilizing to avoid hydration with a hypochlorite solution? 

2. When mashing the chickpeas with a mortar and pestle to simulate the mechanical chewing, were they mixed with salivary fluids? 

3. If the authors are aware of the limitations of the Lowry method, why use this method instead of another one like the Kjeldhal or the Bradford method?

4. How could the authors improve the gastric hydrolysis of germinated chickpeas? Would this affect digestibility and nutrient absorption, and bioavailability?

Author Response

The authors present a study about the effect of germination on the gastrointestinal digestion and antioxidant activity of chickpeas. 

The manuscript is well organized, and the results are interesting. The study is potentially helpful for the food industry and consumers.

  1. Does soaking the dried chickpeas in a hypochlorite solution not affect the germination? Shouldn't the chickpeas be rehydrated before sterilizing to avoid hydration with a hypochlorite solution?

Dry chickpeas were soaked in a 2% (v/v) 0.5 M sodium hypochlorite solution for 10 minutes to sterilize, then washed and soaked overnight. We did not see any significant effect on germination efficiency in this process.

  1. When mashing the chickpeas with a mortar and pestle to simulate the mechanical chewing, were they mixed with salivary fluids? 

Yes, it happened sequentially, then adding salivary fluid, and mashing again (to obtain smaller particle size).

  1. If the authors are aware of the limitations of the Lowry method, why use this method instead of another one like the Kjeldhal or the Bradford method?

Interesting point, as we did use the peptide assay as a complementary and more focus on that, we did not use other total protein determination methods like Kjeldhal or the Bradford.

  1. How could the authors improve the gastric hydrolysis of germinated chickpeas? Would this affect digestibility and nutrient absorption, and bioavailability?

Our data indicate (Fig 3A, B, C), that digestibility is increased after germination, and it could modulate the absorption and bioavailability, but in this study, we focus more on bioactivity rather than absorption and bioavailability. Future studies should evaluate the effect of germination in modulating absorption and bioavailability.

Round 2

Reviewer 1 Report

In the Manuscript " Germination and Simulated Gastrointestinal Digestion of  Chickpea (Cicer arietinum L.) in Exhibiting in-vitro Antioxidant Activity in Gastrointestinal Epithelial Cells" the aim of the authors was to determine whether the impact of germination and gastrointestinal digestion on the release of dietary compounds with antioxidant activity from chickpeas (Cicer arietinum L.). The paper is a potentially interesting manuscript that necessity of same integrations.

1)      The authors should follow the antioxidant activity of the Chickpeas germinating (CG) and GI samples also in a confluent Caco-2 cellular system, that is like to enterocyte, in which is possible induced oxidative stress.

2)      It would also be interesting to show whether the antioxidant effect of CG and GI extracts also influenced the adjacent cell-cell junctions in vitro.

Author Response

Thank you, the reviewer, for asking the same questions again.

  1. The authors should follow the antioxidant activity of the Chickpeas germinating (CG) and GI samples also in a confluent Caco-2 cellular system, that is like to enterocyte, in which oxidative stress has been induced.

As mentioned earlier, we aimed to perform the assay on the Caco2 cells, but the signals of the antioxidant assay were NOT stable and strong when we use the Caco2 Cells; on the other hand, when we use the HT-29 cells, the signals were stable and strong, which ensure reproducible results. You will find that people have performed the same assay using Caco2 cells, but we did not get a stable result, so we went for a different option. Yes, Caco2 is classified as enterocytes, but our goal was to look for it in an Intestinal cells model; the first type of cells with the germinated digesta will interact. In our future studies, we aim to use other regular gastrointestinal cells to see the broad impact of our germinated chickpea.

2. It would also be interesting to show whether the antioxidant effect of CG and GI extracts also influenced the adjacent cell-cell junctions in vitro.

Thank you for the comment; in our future study, we are focusing on cell-cell junctions and protein expression after our treatments; adding those data will require additional time and resources, which is out of the scope of this manuscript.